# Viewing-Angle-Enhanced and Dual-View Compatible Integral Imaging 3D Display Based on a Dual Pinhole Array

**DOI:** 10.3390/mi15030381

**Published:** 2024-03-13

**Authors:** Hui Deng, Guojiao Lv, Huan Deng, Zesheng Liu

**Affiliations:** 1School of Electronic Engineering, Chengdu Technological University, Chengdu 611730, China; denghui@cdtu.edu.cn (H.D.);; 2College of Electronics and Information Engineering, Sichuan University, Chengdu 610065, China

**Keywords:** integral imaging (InIm), dual-view 3D display, viewing angle enhancement, dual pinhole array

## Abstract

Conventional integral imaging (InIm) three-dimensional (3D) display has the defect of a small viewing angle and usually presents a single 3D image. In this paper, we propose a viewing-angle-enhanced and dual-view compatible InIm 3D display system. The crosstalk pixel areas within the conventional elemental images (EIs) that result in image crosstalk were effectively utilized either for viewing angle enhancement or for dual-view 3D display. In the viewing-angle-enhanced 3D display mode, a composite elemental image (CEI) that consisted of a normal EI and two view-enhanced EIs was imaged by a dual pinhole array and formed an extended 3D viewing area. A precisely designed mask array was introduced to block the overlapped rays between adjacent viewing areas to eliminate image crosstalk. While in the dual-view 3D display mode, a CEI was composed of image information of two different 3D scenes. With the help of the dual pinhole array and mask array, two different 3D images were reconstructed for the left and right perspectives. Experiments demonstrated that both the left and right sides were increased by 6 degrees from the conventional 3D viewing angle, and also, a dual-view 3D display effect that retains the same viewing angle as the conventional system was achieved. The proposed system has a compact structure and can be freely switched between two display modes.

## 1. Introduction

Integral imaging (InIm) is a light-field three-dimensional (3D) display technology that uses a lens array to record and reconstruct a 3D scene. Since firstly proposed by Lippmann, InIm has attracted great attentions because of its merits such as continuous viewpoints, visual-aid-free, and no visual fatigues [1,2,3,4,5,6]. However, orthoscopic 3D images can be observed only in a narrow angular area where the imaging areas of all the elemental images (EIs) overlap each other, resulting in a small viewing angle. Furtherly, only part of the pixels within an EI contribute to the viewing angle, named here as the effective pixel area, and the rest are related to the image crosstalk, named as the crosstalk pixel area. This means a waste of the limited display information [7,8,9]. Many efforts have been devoted to addressing these issues. For example, Kim et al. used a curved lens array to realize a convergent optical structure and increase the public viewing zones of the display system [10,11]. They also used a diverging light source to illuminate an SLM with the modulation of a lens array, and a virtual 3D image was formed behind the display screen to achieve viewing-angle-enhanced InIm [12]. Furthermore, Park, G. et al. proposed a method based on head tracking to improve the viewing angle and viewing distance in the InIm system [13]. Wang et al. proposed an InIm-based tabletop light-field 3D display with a compound lens array and achieved a radial viewing angle of 68.7° on a 43.5-inch 3D display [14]. Sang et al. presented a time-multiplexed light-field display with up to a 120° wide viewing angle by using three groups of directional backlights and a fast-switching liquid crystal display (LCD) panel [15]. These methods significantly increased the viewing angle, but the system structures are too complex to be commercially applied. Deng et al. proposed a micro-image array distribution method with a convergent structure that extended the viewing angle by appropriately enlarging the pitch of elemental images. This method can effectively eliminate image crosstalk and expand the viewing angle, but it is suitable for a specified optimum viewing distance, and image scaling based on interpolation in this method may cause 3D image distortion [16]. In our previous study, we had analyzed the contributions of each EI to the public viewing zone, and deduced the distributions of effective pixel area that contribute to the orthoscopic 3D images, as well as the crosstalk pixel area corresponding to the image crosstalk. By filling the crosstalk pixel area with correct image contents, image crosstalk can be eliminated and therefore the viewing angle can be enlarged [8].

On the other hand, dual-view display has appeared as a new display trend in recent years and has attracted great interest. It presents two different images on the same display screen and makes each viewer observe a specified image from a different viewing direction. For example, in the vehicle display, the dual-view display technology can provide navigation information for the driver on one side, and at the same time provide entertainment for the passenger on the other side. During medical surgical operation, with the help of dual-view display technology, the doctor can see the patient’s status information during the operation, while the assistant can also check the patient’s historical data and other information simultaneously [17,18,19]. Some researchers have proposed interesting 3D displays by combining dual-view display with InIm 3D display technology and realized a dual-view glasses-free 3D display effect. Wu et al. divided each element image (EI) into left and right groups, which display different contents, respectively, and present different 3D images in the left and right viewing directions simultaneously. However, this method leads to the problem of small viewing area, and there is crosstalk between the two viewing areas [20,21]. In 2016, Wang and Deng et. al. proposed a time-multiplexing switching scheme by using an orthogonal polarizer array and a polarization switcher to match the different scenes, so as to realize dual-view integral imaging 3D display [22]. They also designed a mask plate with a time-multiplexing switch to block the imaging of adjacent lenses, and matched with the conversion of different display modes to achieve a large viewing angle and high-spatial-resolution dual-view integral imaging display [23]. The above two schemes combine with time-multiplexing technology to improve the spatial resolution of dual-view integral imaging at the cost of losing the time resolution. However, the need for synchronous switching of the display system leads to a complex system structure, and there is still overlap in the viewing area. Another method, which uses the refraction effect of a prism array to separate the two viewing areas, can effectively eliminate the overlap of the two viewing areas and solve the crosstalk problem [24]. However, the device requires a point light source illumination and collimation with Fresnel lens, and the prism array itself is thick (13 mm), which increases the thickness of the system by several times.

In this paper, we present a viewing-angle-enhanced and dual-view compatible InIm 3D display system. In the proposed system, the crosstalk pixel area can be utilized either to display view enhancement information to extend the viewing angle, or to display another 3D scene to realize a dual-view 3D display. The proposed system has a compact structure and can be freely switched between two display modes, and also the pixel utilization of the display screen is obviously increased.

## 2. Principles

The conventional integral imaging structure, as shown in Figure 1, consists of a two-dimensional (2D) display screen and a pinhole array (or a microlens array). The 2D screen displays an elemental image array (EIA) composed of many elemental images (EIs), and each EI is imaged by a corresponding pinhole in front of it to reproduce the 3D image. The common area of each EI through the pinhole is the stereoscopic viewing area. Taking the horizontal direction as an example, as shown in the yellow area in Figure 1, the horizontal width of the stereoscopic viewing area is *D*, and the angle generated by the stereoscopic viewing area and the center of the pinhole array is named as 3D viewing angle *θ*.

A correct 3D image can be viewed within the stereoscopic viewing area, and there is a crosstalk area outside the stereoscopic viewing area, as shown in the gray area in Figure 1. In the crosstalk area, the imaging lights are aliasing between two EIs, which seriously affects the stereoscopic viewing effect.

It is found that the effective information observed in the stereoscopic viewing area corresponds to the specific area of each EI, which is called the effective pixel area, as shown in the red area of the EIA in Figure 1, and the width of the effective pixel area *E* of each EI is:(1)$$E=p-\frac{\left(M-1\right)pg}{L}$$
where *p* is the unit pitch of the pinhole array, *g* is the gap between the pinhole array and the 2D display screen, *L* is the viewing distance, and *M* represents the number of EIs in the horizontal direction.

The position of the effective pixel in each EI moves evenly starting from the first EI on the left edge to the last EI on the right edge. The amount of movement $∆m_{i,j}$ of the effective pixel in the horizontal direction of EI in row *i* and column *j* is:(2)$$∆m_{i,j}=\frac{\left(j-1\right)pg}{L}$$

The area that produces a crosstalk image between two adjacent effective pixel area is defined as the crosstalk pixel area, and its width *C* is:(3)$$C=\frac{\left(M-1\right)pg}{L}+\frac{pg}{L}=\frac{Mpg}{L},$$

Due to the existence of the crosstalk pixel area, the display information of the conventional integral imaging structure is not fully utilized. Moreover, overlapping stereoscopic viewing areas and crosstalk will seriously affect the viewing experience of viewers.

As shown in Figure 1, the viewing angle of conventional structural *θ* should be:(4)$$\theta =2\arctan\left(\frac{p}{2g}-\frac{\left(M-1\right)p}{2L}\right)$$

If the crosstalk pixel area is fully utilized, the viewing angle can be maximized to:(5)$$\theta_{Max}=2\arctan\left(\frac{p}{2g}+\frac{\left(M-1\right)p}{2L}\right)$$

Ref. [16] proposed a structure that achieves $\theta_{Max}$ at the optimal viewing distance.

In order to effectively utilize the crosstalk pixel area and improve the information utilization of the display screen, this paper proposed an integral imaging structure based on a dual pinhole array, and the proposed structure is compatible with both viewing angle enhancement and dual view feature. The schematic diagram of the proposed viewing-angle-enhanced and dual-view compatible integral imaging 3D display system is shown in Figure 2. It mainly consists of a two-dimensional (2D) display panel, a mask array, and a dual pinhole array. A series of composite elemental images (CEIs) is displayed on the 2D display panel. In the viewing-angle-enhanced mode, the CEI contains a normal EI and two view-enhanced EIs. While in the dual-view 3D display mode, the CEI is composed of two EIs corresponding to different scenes. The dual pinhole array has two small holes in each cell, each of which penetrates rays from a certain area of the CEI to form an enhanced viewing area or dual 3D viewing areas in the viewing space. The mask array located between the 2D display panel and the dual pinhole array blocks rays that produce overlaps in the 3D viewing area to eliminate image crosstalk. By setting two pinholes in each cell, the crosstalk pixel areas in the conventional InIm system are effectively utilized for either viewing angle enhancement or dual-view 3D display.

The viewing-angle-enhanced mode is taken as an example to analyze the working principle of the proposed system. In a CEI, the pixel area filled in yellow represents a normal EI, which corresponds to the effective pixel area in the conventional InIm display, as shown in Figure 3a, and the pixel areas filled in blue and green denote the view enhancement information for the left and right viewing zones, respectively. The structure of the mask array is shown in Figure 3b, where the black part blocks the rays and the white part allows rays to pass through. Each unit of the dual pinhole array contains two pinholes, pinhole 1 and pinhole 2, as marked in yellow and green, respectively, in Figure 3c. The CEIs, the mask array and the dual pinhole array have the same number of units.

Figure 4a shows the 3D viewing area of the proposed system formed by the normal EIs. Take the *i*^th^ unit for example. The normal EI, the left view-enhanced EI, and the right view-enhanced EI in the *i*^th^ CEI are denoted as NEI*_i_*, LEI*_i_*, and REI*_i_*, respectively. Two pinholes within the *i*^th^ unit of the dual pinhole array are denoted as P^1^*_i_* and P^2^*_i_*, respectively. Rays emitted by NEI*_i_* pass through P^1^*_i_* and P^2^*_i_* and form two 3D viewing areas, as denoted by *D*_N_. The formed two 3D viewing areas *D*_N_ have equal width and are symmetrically distributed with the central axis of the display screen. However, the rays pass through pinhole P^2^*_i_*_-1_ of the former unit and pinhole P^1^*_i_*_+1_ of the latter unit and will overlap with that coming from the view-enhanced EIs and cause crosstalk. Hence, a mask array is introduced to block these rays.

The 3D viewing areas formed by two view-enhanced EIs are shown in Figure 4b. Rays emitting from LEI*_i_* pass through pinhole P^2^*_i_*_+1_ of the latter unit and form an additional 3D viewing area, *D*_LE_, on the left side of *D*_N_. Similarly, another 3D viewing area, *D*_RE_, is formed on the right side of *D*_N_ by the rays coming from REI*_i_* and passing through P^1^*_i_*. The rest of the rays coming from LEI*_i_* and REI*_i_* and passing through the adjacent pinholes are effectively blocked by the mask array to avoid image crosstalk.

Figure 5 shows a complete viewing area distribution of the proposed system working in viewing-angle-enhanced 3D display mode. Rays emitted from the normal EIs and the view-enhanced EIs pass through the pinholes and form two enhanced 3D viewing areas which are *D*_LE_ + *D*_N_ on the left side, and *D*_N_ + *D*_RE_ on the right side, respectively. With the help of the mask array, stray lights that cause crosstalk are blocked so that the viewing areas do not overlap with each other.

To effectively block the crosstalk rays, the mask array has to be properly designed. The widths of the normal 3D viewing area, *D*_N_, and the enhanced 3D viewing areas, *D*_LE_ and *D*_RE_, can be expressed as:(6)$$D_{N}=\frac{p}{g}\left[L-\left(M-1\right)g\right],$$
(7)$$D_{\mathrm{LE}}=D_{\mathrm{RE}}=\frac{p}{2g}\left[L-\left(M-1\right)g\right].$$

In CEI, the width of NEI, *m*, is equal to the sum of the widths of LEI and REI:(8)$$m=\frac{Mp}{2M-1}.$$

The distance *l* between the mask array and the 2D display screen should be:(9)$$l=\frac{2mg}{\frac{p}{2}+2m},$$

Furthermore, the width of the light-blocking area *b* is equal to that of the light-transmitting area *w*.
(10)$$b=w=3\frac{m\left(g-l\right)}{g}=\frac{3pm}{p+4m}$$

The 3D viewing angle, *θ_S_*, of the proposed system, and that of the conventional InIm system, *θ_N_*, can be expressed by Equations (9) and (10). Comparing the two equations, the 3D viewing angle of the proposed system is obviously improved.
(11)$$\theta_{S}=\arctan\left[\frac{3p}{2}\left(\frac{1}{g}-\frac{M-1}{L}\right)\right],$$
(12)$$\theta_{N}=2\arctan\left[\frac{p}{2}\left(\frac{1}{g}-\frac{M-1}{L}\right)\right].$$

When the proposed system works in the dual-view 3D display mode, as shown in Figure 6, the information of two different 3D scenes is loaded into EI 1 and EI 2, respectively. EI 1 is assigned with the same pixel area as the normal EI shown in Figure 5, and EI 2 is assigned with the same pixel area as the two-view-enhanced EI. In the *i*^th^ unit, rays of EI^1^*_i_* pass through pinholes P^1^*_i_* and P^2^*_i_* and form two viewing areas for 3D image 1, while the rays of EI^2^*_i_* pass through pinhole P^1^*_i_* and pinhole P^2^*_i+2_* and form two viewing areas for 3D image 2. The overlapped rays of the two EIs are blocked by the mask array to avoid image crosstalk. Therefore, two different 3D images can be observed from different perspectives.

As shown in Figure 6, in dual-view 3D display mode of the proposed system, the 3D image 1 is repeated twice in the center position, while the 3D image 2 is presented on the sides, which is suitable for the situation when there is more viewing need for one scene, or there is primary information and secondary information. The primary information is presented in the middle, which can be viewed by multiple people, while the secondary information is presented on the sides, which is viewed by fewer people. This mode has a more flexible dual-view display effect.

Therefore, with the help of the mask array and dual-pinhole array set in the proposed system, the crosstalk pixels are effectively utilized to either extend the 3D viewing angles or to realize a dual-view 3D display.

## 3. Experiments and Results

In the experiment, we developed a prototype to verify the feasibility of the proposed viewing-angle-enhanced and dual-view compatible InIm 3D display system. For comparison, we also built up a conventional InIm display. Both systems used the same 2D display panel (the AOC U2790PQU) with a 4K resolution and a pixel size of 0.155 mm. The mask array was made by attaching a laser-printed high-resolution film (printed by DX4600II) to a high-transmittance optical substrate, and the dual pinhole array was made by the same method. The detailed specifications and parameters of the conventional system and proposed system are listed in Table 1. Two 3D models, one a car, and the other a body model, are created in the software 3ds Max 2021 to represent 3D scene1 and 3D scene 2, respectively, as shown in Figure 7a,b. EIs for the conventional InIm system, and two sets of CEIs for the proposed system working in the viewing-angle-enhanced mode and the dual-view 3D display mode, respectively, are generated, as shown in Figure 7c–e. A digital camera is used to capture the reconstructed 3D images from different perspectives. In Table 1, we also set up a set of theoretical data of a wide viewing angle system described in Ref. [16] for comparison.

For the conventional InIm display, the viewer located at the viewing distance of 1500 mm can observe a complete 3D image of the “car” in a continuous angular range from −11° to 11°. Figure 8b–d show three perspective images captured by the camera. However, 3D images observed beyond this angular range show obvious ghost images and cracks, as shown in Figure 8a,e which are captured from −13° and 13°, respectively. This means the conventional InIm system has a viewing angle about 22°. For the proposed system, orthoscopic 3D images can be viewed in a continuous angular range from −30° to −1° and from 1° to 30°. Figure 8 shows eight perspective images. Among them, Figure 9c–f show basically the same display results as in Figure 8b,d, indicating that the imaging rays in the conventional effective pixel area of the proposed system are barely affected. Meanwhile, Figure 9a,b show the right viewing-angle-enhanced image of the “car” and Figure 9g,h show the left viewing-angle-enhanced image. The experimental results show that the proposed system increased the 3D viewing angle from 22° to 30°. Experimental results show that the viewing angle of the proposed system is increased by 8° on both sides of the viewing angle of the conventional InIm system. Although the single viewing angle of the proposed system can only reach 30°, it is less than the theoretical value of 38.7° that can be reached in the reference [16], but the proposed system provides the left and right viewing angles, respectively, and both the left and right viewing angles can reach 30°, compared with the conventional structure, and the viewing angle is increased by 16° in total, which is better than the viewing angle increase effect of the reference [16].

In the dual-view 3D display experiment, the structural parameters are consistent with the viewing-angle-enhanced mode, the NEI pixels of the CEI displays stereo scene 1, and the REI+LEI part displays stereo scene 2. The dual-view 3D display results are shown in Figure 10, where the “model” is viewed from −44° to 24°, and the “car” is viewed from −22° to −1°. The presented 3D images are complete and clear without crosstalk.

In traditional dual-view display, the viewing areas of two scenes are repeated alternately, while in the proposed structure, the viewing areas of scene 1 will repeat twice as often as the viewing area of scene 2, which can be applied to the situation where there is more viewing demand for scene 1. For example, in surgery, the current lesion information of the patient, as the primary information, needs to be presented to the attending physicians, while the data such as blood pressure and heartbeat, as auxiliary information, need to be provided to the assistants for real-time monitoring. The primary information is presented in the middle, which is viewed by more people, while the auxiliary information is presented on the sides, which is viewed by fewer people.

In addition, in the proposed structure, the mask array and pinhole array seemingly will block most of the light, but in fact, the light blocked by the mask array is cross-talk light which forms the wrong visual effect, and the effective imaging light passes through the transmittance area. That is, the mask array has no influence on the optical efficiency of the correct 3D image. Compared with the traditional integral imaging display structure with a single pinhole array, the optical efficiency of the proposed structure is doubled by setting two pinholes in each unit.

Ultimately, the above experimental results show that the proposed system can effectively use the crosstalk pixel area to display the left and right viewing-enhanced information on the basis of retaining the 3D viewing effect of the conventional system, and obtain a larger view angle. At the same time, it is compatible with the dual-view 3D display. The proposed system can meet the needs of large viewing angle, and multi-scene and multi-region viewing. The design of the mask array effectively blocks the crosstalk rays and eliminates the crosstalk image in the 3D viewing angle.

## 4. Conclusions

The existence of crosstalk pixels in the EIs of conventional InIm results in a waste of display information and an uncomfortable 3D viewing effect. To address this issue, we proposed a viewing-angle-enhanced and dual-view compatible InIm 3D display system. It consists of a 2D display screen, a dual pinhole array, and a mask array. In the proposed system, the crosstalk pixel areas could be effectively utilized for displaying the view enhancement information for the left and right perspectives of a 3D scene to extend the viewing angle on both sides of the conventional InIm 3D viewing area, or for displaying the image content of another 3D scene to realize a dual-view 3D display. By appropriately designing the parameters of the CEI, the dual pinhole array, and the mask array, the rays that lead to image crosstalk are effectively blocked. Experiments demonstrated that the proposed system can present a viewing-angle-enhanced or dual-view 3D display effect. Two display modes can be freely switched without any changes to the system parameters. The proposed system makes efficient use of the pixels of the 2D display and meets diverse viewing requirements flexibly.

## Figures and Tables

**Figure 1 micromachines-15-00381-f001:**
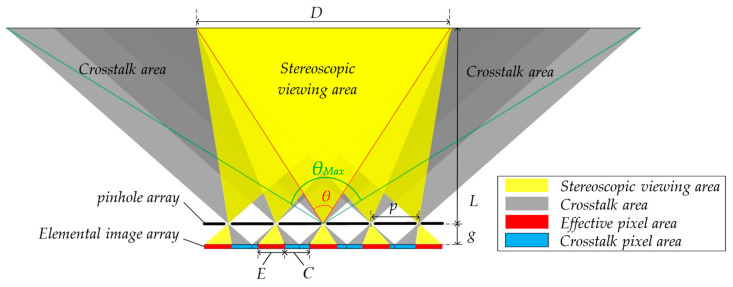
Distribution of effective pixel area and crosstalk area of EIA.

**Figure 2 micromachines-15-00381-f002:**
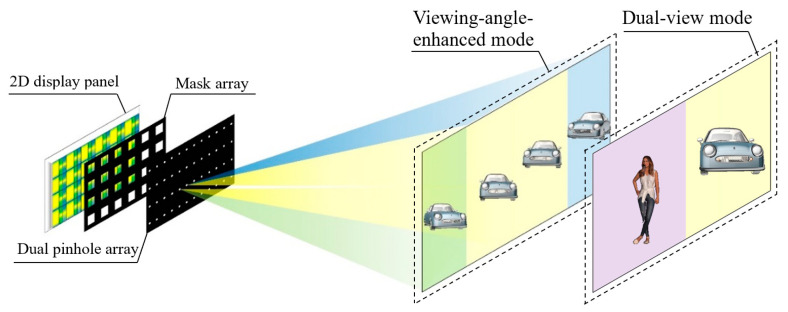
Schematic diagram of the proposed viewing-angle-enhanced and dual-view compatible integral imaging three-dimensional (3D) display based on a dual pinhole array.

**Figure 3 micromachines-15-00381-f003:**
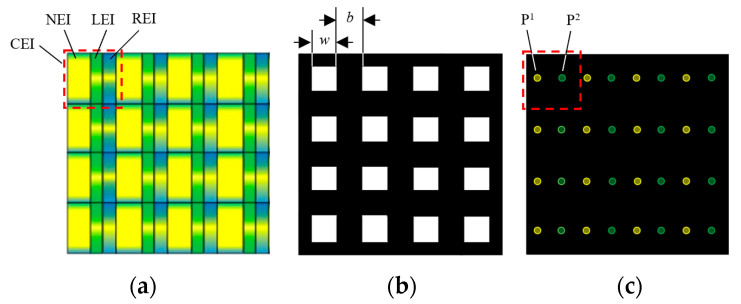
Structure diagram of (**a**) CEIs, (**b**) mask array, and (**c**) dual pinhole array.

**Figure 4 micromachines-15-00381-f004:**
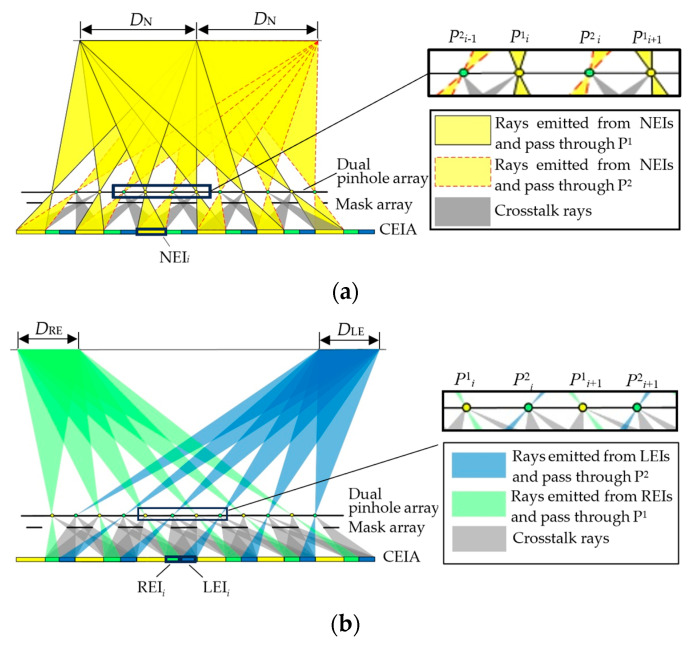
Three-dimensional viewing area distribution of (**a**) normal EIs and (**b**) view-enhanced EIs.

**Figure 5 micromachines-15-00381-f005:**
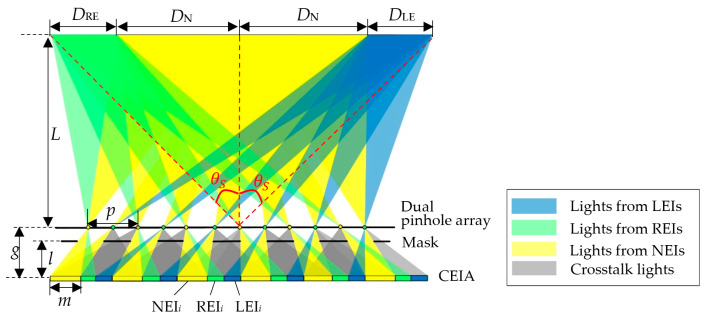
Complete 3D viewing area distribution of the proposed system in viewing-angle-enhanced 3D display mode.

**Figure 6 micromachines-15-00381-f006:**
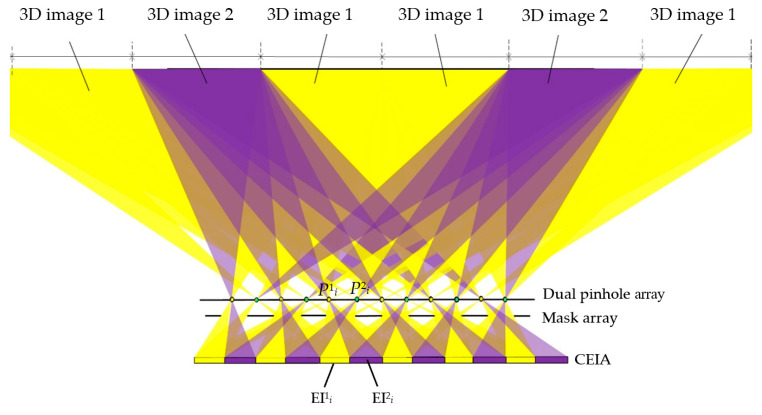
Three-dimensional viewing area distribution of the proposed system in dual-view 3D display mode.

**Figure 7 micromachines-15-00381-f007:**
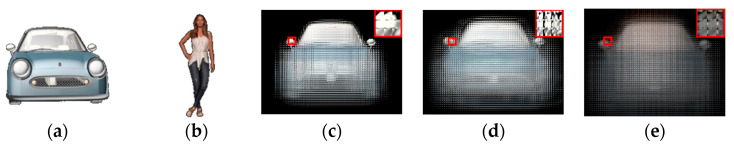
(**a**) Three-dimensional scene 1 and (**b**) 3D scene 2 created in software 3ds Max 2021, (**c**) EIs for conventional InIm display, (**d**) CEIs for viewing-angle-enhanced mode of 3D scene 1, and (**e**) CEIs for dual-view mode of 3D scene 1 and 3D scene 2.

**Figure 8 micromachines-15-00381-f008:**
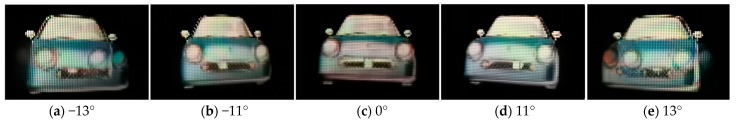
Three-dimensional display results of the conventional system at different angles.

**Figure 9 micromachines-15-00381-f009:**
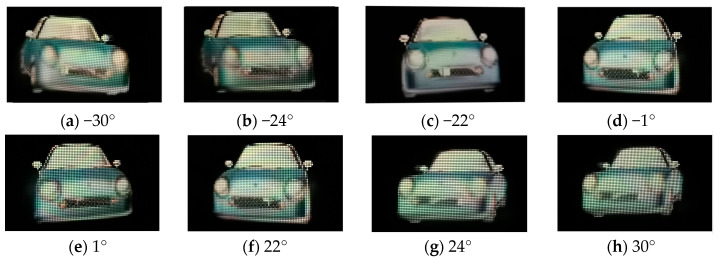
Three-dimensional display results of our proposed system at different angles.

**Figure 10 micromachines-15-00381-f010:**
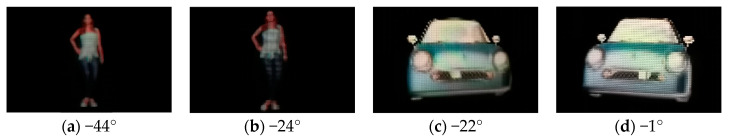
Dual-view 3D display results of the proposed system.

**Table 1 micromachines-15-00381-t001:** The parameters of the experimental systems.

Components	Specifications	Viewing-Angle-Enhanced Mode of Proposed System	Dual-View Mode of Proposed System	Conventional InIm System	Wide Viewing Angle System in Reference [16]
2D display screen	Model	AOC U2790PQU	AOC U2790PQU	AOC U2790PQU	AOC U2790PQU
	Size	27 inches	27 inches	27 inches	27 inches
	Resolution	3840 × 2160 pixels	3840 × 2160 pixels	3840 × 2160 pixels	3840 × 2160 pixels
	Pixel pitch	0.155 mm	0.155 mm	0.155 mm	0.155 mm
EIs	Resolution of LEI	6 × 24 pixels	6 × 24 pixels	N/A	N/A
	Resolution of REI	6 × 24 pixels	6 × 24 pixels	N/A	N/A
	Resolution of NEI	12 × 24 pixels	12 × 24 pixels	24 × 24 pixels	24 × 24 pixels
	Number of EIs	160 × 90	160 × 90	160 × 90	160 × 90
	Pitch $p_{EI}$	3.73 mm	3.73 mm	3.73 mm	3.73 mm
Pinhole array	Pitch p	3.718 mm	3.718 mm	3.73 mm	3.718 mm
	Gap *g*	4.7 mm	4.7 mm	4.7 mm	4.7 mm
Mask array	Black area *w*	1.86 mm	1.86 mm	N/A	N/A
	White area *b*	1.86 mm	1.86 mm	N/A	N/A
	Gap between mask array and CEIs *l*	3.1 mm	3.1 mm	N/A	N/A
Viewing area	Distance *L*	1500 mm	1500 mm	1500 mm	1500 mm
	3D viewing angle	*θs* = 30.83°	$\theta_{N}=$ 22.5°	$\theta_{N}=$ 22.5°	$\theta_{i}=$ 38.7°

## Data Availability

Data is contained within the article.

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
