# Peer review of "Viewing-Angle-Enhanced and Dual-View Compatible Integral Imaging 3D Display Based on a Dual Pinhole Array"

_micromachines, 2024, doi:10.3390/mi15030381_

Round 1

Reviewer 1 Report

Comments and Suggestions for Authors

Comments on the Quality of English Language

Reviewer 2 Report

Comments and Suggestions for Authors

I feel it is a very nice piece of work, with very clear presentation of the proposed schemes and the experimental results to verify its effectiveness. The principles and results have been clearly described and presented, showing a 3D display system with enhanced viewing angle and a dual-view display system.

 I just have some minor questions, e.g.:

 1)     What is the accuracy/tolerance do you need for the mask array and pinholes ?

2)     The image shown in 8(a) is less clear, which corresponds to large viewing angle of negative 30 degree ? Can you improve the image quality further ?

As mentioned above, I feel the paper is very clear in its presentation.

Reviewer 3 Report

Comments and Suggestions for Authors

In this manuscript, the authors use a pinhole array and a mask to enlarge the viewing angle of the InI display. The concept is interesting and there are several questions:

1) Efficiency: What is the system efficiency? The pinhole array and the mask block most of the light and will dramatically reduce the system's efficiency. Please add a section to discuss the efficiency loss and the corresponding brightness.

2) For Figures 7 and 8, the "screen door" on the images is a moiré pattern during the shooting process or a defect in the image itself? Please explain if it is caused by the display system.

Round 2

Reviewer 1 Report

Comments and Suggestions for Authors

The authors did not answer to most of my comments. Particularly, 

1. The definition is not given.

2. The angles are not calculated.

5. The vertical distortions are not considered.

6. No column is added.

7. The reason is not specified.

Round 3

Reviewer 1 Report

Comments and Suggestions for Authors

The authors responded to most of the remaining comments. However, authors should keep in mind that the reviewer and readers should understand the unclear points that require explanation.

1. Definition.

OK. The definition is still unclear, but acceptable.

However, authors should be aware that crosstalk is produced from almost all areas, and this should be mentioned in the text of the article. Please refer to [Appl. Opt. 35(10), 1705-1710, 1996], [Proc. SPIE 6778, 677804, 2007], [J. Disp. Technol. 4(1), 109-114, 2008], [OSA Tech. Digest (DH and 3D Imaging), DWB27, 2009], [IMID Tech. Digest, 1014-1017, 2009] and the like.

2. Angles.

OK, but the formula is not included in the text of the paper.

5. Vertical distortions.

OK.

6. Column.

The column for dual-view is added.

However, a column was also asked to compare with the conventional system with a regular (not double) number of pinholes, the angle of which is 38° (as calculated by the authors in response to Note 2) is not yet included in the Table.

7. Reason.

OK, but the illustration and explanation are not included in the text.
